# Is There a Place for Lewy Bodies before and beyond Alpha-Synuclein Accumulation? Provocative Issues in Need of Solid Explanations

**DOI:** 10.3390/ijms25073929

**Published:** 2024-04-01

**Authors:** Paola Lenzi, Gloria Lazzeri, Michela Ferrucci, Marco Scotto, Alessandro Frati, Stefano Puglisi-Allegra, Carla Letizia Busceti, Francesco Fornai

**Affiliations:** 1Human Anatomy, Department of Translational Research and New Technologies in Medicine and Surgery, University of Pisa, 56126 Pisa, Italy; paola.lenzi@unipi.it (P.L.); gloria.lazzeri@unipi.it (G.L.); michela.ferrucci@unipi.it (M.F.); m.scotto5@studenti.unipi.it (M.S.); 2IRCCS—Istituto di Ricovero e Cura a Carattere Scientifico, Neuromed, 86077 Pozzili, Italy or alessandro.frati@uniroma1.it (A.F.); stefano.puglisiallegra@neuromed.it (S.P.-A.); carla.busceti@neuromed.it (C.L.B.); 3Neurosurgery Division, Department of Human Neurosciences, Sapienza University, 00135 Roma, Italy

**Keywords:** sequestosome (p62), poly-ubiquitin, phagophore, endosome, multivesicular bodies, retromer, lysosome, autophagosome, autophagoproteasosome, mitochondria

## Abstract

In the last two decades, alpha-synuclein (alpha-syn) assumed a prominent role as a major component and seeding structure of Lewy bodies (LBs). This concept is driving ongoing research on the pathophysiology of Parkinson’s disease (PD). In line with this, alpha-syn is considered to be the guilty protein in the disease process, and it may be targeted through precision medicine to modify disease progression. Therefore, designing specific tools to block the aggregation and spreading of alpha-syn represents a major effort in the development of disease-modifying therapies in PD. The present article analyzes concrete evidence about the significance of alpha-syn within LBs. In this effort, some dogmas are challenged. This concerns the question of whether alpha-syn is more abundant compared with other proteins within LBs. Again, the occurrence of alpha-syn compared with non-protein constituents is scrutinized. Finally, the prominent role of alpha-syn in seeding LBs as the guilty structure causing PD is questioned. These revisited concepts may be helpful in the process of validating which proteins, organelles, and pathways are likely to be involved in the damage to meso-striatal dopamine neurons and other brain regions involved in PD.

## 1. Introduction

### 1.1. Is Alpha-Synuclein the Major Component of LBs?

When reading either a review or research article dealing with alpha-synuclein (alpha-syn) and/or neuronal inclusions (i.e., Lewy bodies, LBs) in Parkinson’s disease (PD), some classic statements are often taken for granted. This is generally expressed by the following sentence: “Lewy bodies are neuronal inclusions, where the major structural component is α-synuclein”. Such a concept also applies when primarily referring to alpha-syn, which is usually introduced by the following sentence: “α-synuclein represents the major component of Lewy bodies”. These strong concepts arise from a dogmatic belief which assumes that light microscopy allows for the reliable establishment of the exact amount of alpha-syn within LBs. Indeed, quantitative studies of LBs’ components in situ and quantitative comparisons between LBs’ components based on in situ stoichiometry are still lacking. The current beliefs derived from alpha-syn immunohistochemistry are presently under debate. In fact, some of the recent literature suggests a structure of LBs which is way more complex than alpha-syn aggregates [1,2]. This novel evidence challenges routine statements about a quantitative predominance of alpha-syn. The prevalence of this protein and the relative amounts of protein vs. non-protein components within LB structures are presently questioned. In fact, LBs now appear to contain a significant amount of non-protein cell components made up of lipid-rich membranous structures. These mostly consist of vesicles, including endosomes, autophagosomes, lysosomes, and mitochondria [3,4,5,6,7,8,9], where the protein density from electron micrographs seems to be negligible [1]. This leads to toning down the definition of LBs as proteinaceous inclusions (independently from the relative amount of alpha-syn compared with other proteins occurring within LBs). Thus, the experimental mimicking of LBs by injecting the cells with alpha-syn aggregates is unlikely to reproduce bona fide LBs, being, at most, an extreme attempt to replicate just one of the many components of LBs. In fact, other proteins and lipid vesicles are abundant within LBs. This novel scenario does not rule out the role of alpha-syn in forming LBs, it rather calls for implementing our knowledge about LBs’ structure. In fact, the presence of remarkable amounts of alpha-syn, as shown by light microscopy, is evident and non-debatable. Nonetheless, the quantitative prevalence of alpha-syn within LBs remains questionable.

### 1.2. Is Alpha-Syn the Natural Seed of LBs?

The role of alpha-syn is also considered as prominent in the dynamics of LBs’ formation, which is stigmatized by the following routine statement: “α-synuclein is the building block of Lewy bodies”. However, even their molecular mechanisms, which link alpha-syn with the dynamics of LBs’ formation, remain quite elusive. Such an uncertainty is becoming more and more evident in recent studies. This diverts the vision of the pathophysiology of PD from a single-guilty-protein-induced disorder into a more complex neurobiology of disease. In fact, novel vistas provide evidence for a multi-step dysfunction affecting various proteins and non-protein components such as lipids within organelles involved in the activity of major clearing pathways. In turn, the role of alpha-syn may shift from a disease-specific guilty protein into a structure, which is guilty only by association [10]. In fact, according to recent evidence, there is no need to keep alpha-syn fibrils within neurons to fuel the formation of LBs, since this is an autocatalytic process, which may engage organelles in addition to or rather than single proteins. The term autocatalytic refers to the concept that early seeds of LBs further progress into mature LBs autonomously through self-assembly via the interactions of various components due to their chemical properties. In fact, as reported by recent studies [11,12], LB formation progresses up to the catalysis of early seeds, which are formed by various protein aggregates, protein–lipid aggregates, or even aggregates of membrane-bound organelles in the absence of alpha-syn fibrils [11]. Thus, one might assume that non-protein vesicular components may also be able to seed and grow step-by-step for the formation of LBs, as recently figured out by applying predictive mathematical modeling [11,12], simulated in experimental models [13,14], and confirmed in human post mortem studies [1]. The occurrence of abundant membranous organelles within LBs is likely to foster LB formation, even by their interaction with alpha-syn [11,12]. Apart from lipid membranes, other proteins than alpha-syn are now claimed as potential seeds of LBs, which confirms that the catalytic role of alpha-syn as a building block to start neuronal inclusions is debatable [11,15,16]. In fact, although the artificial in tubo or in vitro feeding of alpha-syn may be intended as a bona fide model of LBs [7], in the natural process of LBs’ maturation, key proteins other than alpha-syn are able to trigger the seeding of pale eosinophilic LBs. This is the case of eosinophilic poly-ubiquitin- and p62-positive LBs within damaged nigral DA neurons developing in mice knocked out for alpha-syn [13]. This evidence indicates that the seeding of LBs defined as eosinophilic p62- and poly-ubiquitin-positive inclusions may develop independently from the presence of alpha-syn. In line with this, a recent manuscript by Noda et al. [14] highlights a key role of the homeostatic levels of p62 in triggering LBs’ formation [14]. These data are in line with evidence obtained from mice, where autophagy inhibition was induced by site-specific ATG7 depletion (ATG7 conditional knock outs) within DA neurons of the substantia nigra. These mice develop eosinophilic inclusions, where p62 accumulation precedes by months the presence of alpha-syn [15]. This delayed timing for alpha-syn immunostaining compared with p62-immunostaining within LB-like inclusions has been confirmed for other PD-inducing genes such as parkin [17]. Similarly, when PARK22-linked PD is analyzed in ChCHD2 mutant mice, p62 aggregates initiate LB pathology, which also features alpha-syn [16].

### 1.3. Is Alpha-Syn the Unique Guilty Protein in the Onset and Progression of PD?

Similar to the multiple composition and seeding of LBs, the same plurality applies to the molecular mechanisms involved in the neurobiology of PD. In fact, the presence of proteins other than alpha-syn and the occurrence of non-protein structures within LBs suggest alternative mechanisms that drive cytopathology in PD. It was recently postulated that some early endo-lysosomal dysfunctions characterize the neurobiology of PD [3,4,10]. In line with this, it is likely that dysfunctional membrane-bound organelles are strongly involved in the onset and progression of PD, which recapitulates the steps involved in the formation of LBs. The recruitment of non-protein structures within LB may partly depend on the abnormal accumulation of the protein alpha-syn, which leads to abnormal interactions with an otherwise disrupted endo-lysosomal system [7]. For instance, Tiexeira et al. [10] suggested an abnormal interaction between alpha-syn and the endo-lysosomal system as a key step in the onset and progression of PD. This opinion is also supported by Vidyadhara et al. [4] based on the specific role of alpha-syn as a typical pre-synaptic protein, which may produce monomers and aggregates at crucial synaptic domains by interacting with endosomes and retromers. This is also in line with the evidence that roughly 30% of native alpha-syn is suddenly prone to aggregation [18]. These aggregates recruit a number of pre-synaptic proteins to cast clusters of membranes [18]. In fact, native alpha-syn promotes SNARE-complex assembly along with the clustering of synaptic vesicles, which accumulate at the pre-synaptic active zone [19]. This confirms the hypothesis by Teixeira et al. [10], that the accumulation of monomers and aggregates of alpha-syn may disrupt pre-synaptic vesicles’ trafficking, including docking and recycling. This abnormality would impair retromers and the endo-lysosomal system, mostly by engulfing the autophagy-lysosomal pathway [4,10,20]. Moving from non-protein structures, other proteins than alpha-syn may have a fundamental role in the onset and progression of PD. As reported in the original findings by Komatsu et al. [21], p62 plays a critical role in seeding neuronal inclusions and endogenous p62 is indispensable in order to develop neuronal inclusions when autophagy is impaired. This evidence suggests that, while abnormal membranous structures are constantly present within LBs, the occurrence of alpha-syn may not be indispensable, especially during the early disease stages. This is in line with Komatsu’s findings [21], showing the relative abundance of p62 compared with alpha-syn upon autophagy inhibition within catecholamine cells. Autophagy inhibition may be adopted to mimic bona fide LBs in vitro [22], since it occurs primarily in a number of cases of PD [23,24,25,26,27,28,29,30]. The condensed time interval between autophagy dysfunction and protein accumulation provides a perspective to explain the differential increase between p62 and alpha-syn in short time intervals. This is in line with concepts developed following ex vivo experiments, where eosinophilic inclusions mimicking LBs may develop based on p62 and poly-ubiquitin seeding aggregates in the absence of alpha-syn, which adds to eosinophilic inclusions only at long time intervals (several months later) [15]. A similar phenomenon was recently produced in other experimental conditions following the exposure of catecholamine cells to the neurotoxin methamphetamine [31], which is known to produce neuronal inclusions in rodents [32] and humans [33]. In fact, when methamphetamine is administered to catecholamine cells, eosinophilic inclusions staining for alpha-syn appear. When observed under light microscopy, methamphetamine increases both alpha-syn and p62 immunofluorescence. However, these effects are quite different when the level of analysis goes in depth to visualize the stoichiometry of immuno-gold particles joined with alpha-syn or p62 under transmission electron microscopy (TEM). In this latter case, only p62 produces clusters of proteins, which may be viewed as reminiscent of seeds to form neuronal inclusions. In these experimental conditions, at short time intervals, alpha-syn slightly increases; however, this occurs according to a non-clustered, rather scattered pattern. The amount of p62 and poly-ubiquitin staining of these inclusions exceeds ten-fold the amount of alpha-syn, and they are evident as protein clusters [31]. This long introduction serves to outline some discrepancies, which may be helpful in avoiding over-interpretations about the role of alpha-syn in the formation of LBs. In fact, the present review raises some provocative issues, which need to be considered before assuming LBs as a mere synucleinopathy. 

### 1.4. Is the Endo-Autophago-Lysosomal System Defective in PD?

In line with this, most mutations leading to genetic Parkinsonism are key in affecting clearing pathways. This is the case for proteasome, autophagy, endosomes, and lysosomal activity, as it was postulated for two decades [34,35,36,37,38,39,40] and was recently confirmed [10,20,41]. In keeping with a failure of these clearing systems, abnormal mitochondria and an excess of protein cargoes occur in inherited and in sporadic PD. The earliest insight into the role of altered steps in cell clearance as being determinant for the neurobiology of PD derives from the analysis of inherited Parkinsonism, where various steps of the endo-autophago-lysosomal system are defective. In fact, solid evidence indicates that genetic PD mutations affect the structures of genes, which regulate cell-clearing pathways such as autophagy, proteasome, and lysosome. In turn, these are seminal for mitochondrial turnover and the delivery of protein cargoes [37,40,42]. In fact, genetic defects at specific steps in cell-clearing systems (autophagy, endosomes, and proteasome) were related to the abnormal persistence of altered mitochondria and non-effective removal of protein cargoes (including alpha-syn). Based on the significance of mutated genes in causing inherited PD, the relevance of these interconnected compartments involved in cell clearance emerged in sporadic PD. These compartments include autophagosomes [36,38,43,44,45,46,47], proteasome [34,48], endosomes [49,50], retromers [51,52], lysosomes [53,54,55,56,57], mitochondrial turnover [39,58,59,60,61], and primary protein misfolding [62,63,64].

### 1.5. How to Reconcile Alpha-Syn with Defective Clearing Systems within LBs and PD Progression?

The aggregation of lipid membranous structures may involve abnormal interactions between various proteins, including alpha-syn and the endo-autophago-lysosomal system, leading to the neurobiology of PD, with nigral cell loss, striatal dopamine (DA) denervation, and a concomitant buildup of LBs. Thus, once again, it is relevant to decipher the specific structure and dynamics of LBs to verify such an updated hypothesis designed to comprehend the pathogenesis of PD and disclose disease-modifying treatment and biomarkers of PD [1,10,35,65,66,67]. A crucial issue indicates that there are some rare cases where alpha-syn is likely to be the sole and sufficient protein producing PD. This occurs following rare autosomal causes of PD due either to extra copies of the alpha-syn gene or mutations within the alpha-syn gene. These rare forms of PD may interconnect with a failure of the cell-clearing system induced by mutated alpha-syn. In fact, alpha-syn is a substrate for both proteasome and autophagy and it exerts deleterious effects on these seminal clearing pathways when the gene is mutated or following gene multiplications [68] In fact, evidence has been provided that rare and severe phenotypes of PD are induced by multiplication of the alpha-syn gene [68,69,70], which was shown to impede the natural mechanisms of cell clearance [71]. Similarly, point mutations of alpha-syn may produce various phenotypes (depending on the site of mutation) of PD, being powerful inhibitors of cell-clearing pathways such as proteasome [72,73] and autophagy [47,74,75]. In detail, an overexpression of alpha-syn disrupts the placement and shifting of ATG9 from the trans-Golgi network towards LC3-positive vesicles, which determines a loss of autophagosomes and impairs autophagy. 

This long introduction addresses controversial issues about the role of alpha-syn in LBs and the onset and progression of PD. Such multi-faceted evidence calls for analyzing the seminal data that generated the routine concept of alpha-syn as a major component of LBs, the seeding of LBs, and a severe detrimental protein placed on radiating filaments of LBs. With this aim, the following paragraphs seek to determine how the hegemonic concept of alpha-syn as the major component and building block of LBs in PD is broken down in the current literature. This is discussed trying to check whether a potential discrepancy between the actual evidence and routine interpretations exists, which may lead to non-controlled, inferential statements about the relevance of alpha-syn within LBs and PD. 

## 2. The Starting Evidence Which Led to Define Alpha-Syn as the Major Component of LBs

The classic concept, which leads to state an equivalence between LBs and alpha-syn positive inclusions, is grounded in a vast amount of literature showing the reliable staining of LBs by using antibodies directed against alpha-syn. This evidence started in the August 1997, when the occurrence of alpha-syn within LBs was published [76]. In this pioneering manuscript, the occurrence of alpha-syn was detected as a strong peroxidase-labeled immuno-staining following incubation with primary anti-alpha-syn antibodies of paraffin-embedded tissue sections from the substantia nigra. These tissue sections were obtained from six persons affected by PD and four persons affected by dementia with Lewy bodies (DLB). Samples to be sliced were previously fixed either with formalin or ethanol and they were all obtained from MRC Cambridge Brain Bank. The patients affected by DLB possessed a similar pathology, which was otherwise typical of Alzheimer’s disease (AD). At the time when this study was carried out, LBs were known to possess some neurofilaments and ubiquitin (although the authentic antigen is poly-ubiquitin, as indicated by the previous seminal study of Iwatzubo et al. [77]), which was previously used as a marker to stain LBs. Based on the strong peroxidase-labeled immunostaining, the authors literally hypothesized that “α-synuclein may be the main component of the Lewy body in Parkinson’s disease” [76]. In this study, no quantitative measurement of molecular components was carried out, and we still lack a stoichiometric and quantitative analysis assessing the specific proteins and various molecules that compose LBs.

The antibody used in the pioneer study was highly specific for human alpha-syn and was used at the concentration of 1:200 [76]. As reported by the authors, the strong immunostaining provided by this procedure made it difficult to distinguish between the core and the shell (the central and the peripheral part, respectively, of LBs). Nonetheless, representative pictures from this article report a higher intensity in the center compared with the periphery of the LB. In the very same samples, the staining for ubiquitin revealed a similar amount of stained inclusions. In a following manuscript, authored by the same research group, the occurrence of alpha-syn was found to exceed that of ubiquitin immunostaining [78]. This led the authors to assess that alpha-syn should replace ubiquitin as the gold standard for staining LBs. In this latter study, electron microscopy was used with immuno-gold staining to detect alpha-syn within filaments. However, this was carried out after the centrifugation of gross tissue homogenates and following sarkosyl pre-treatment. Such a process leaves unknown the origins of these filaments and their actual presence in situ within LBs. Again, sarkosyl pre-treatment destroys most native proteins originally present within filaments extracted from gross brain centrifugates. These procedures leave a half-hearted inference about the prominent role of alpha-syn within LBs. This methodological weakness becomes striking considering the impact generated by these data within the literature during the following years.

## 3. Limitations and Pitfalls about Evidence on the Primary Role of Alpha-Syn in LB

These two original manuscripts [76,78] posed the bases for the following assumptions: (i) alpha-syn is the major component of LBs; (ii) alpha-syn represents the seeding structure of LBs, and (iii) alpha-syn is the constituent of radiating filaments within LBs. Indeed, some concerns arise when considering the strength of these assumptions. In fact, in these two original manuscripts, as well as in the bulk of subsequently related research, no quantitative assessments about the amounts of alpha-syn were ever carried out in situ within LBs. Instead, the occurrence of alpha-syn within LBs persisted to be inferred from qualitative light microscopy, which, by definition, does not provide a quantitative assessment of any protein. Therefore, the assumption that alpha-syn is the major component of LBs remains quite arbitrary, without any solid demonstration so far through in situ stoichiometry.

### 3.1. Is Alpha-Syn the Major Component of LB?

The only approach allowing for the measurement of protein amounts through in situ stoichiometry consists of staining neurons with immuno-gold within their specific compartments [79]. Instead, available evidence provides a non-quantitative assessment of alpha-syn immunostaining under light microscopy. Based on these findings, no inference can be drawn about the quantitative prevalence of alpha-syn within LBs compared with that of other proteins. Thus, the conclusion that “staining for α-synuclein is more extensive than staining for ubiquitin” is quite strong, considering that a quantitative measurement of these proteins (including a reliable comparison) is still lacking. Even the prevalence of proteins compared with other structures within LBs cannot be established by these studies. In fact, at present, we still lack solid knowledge about which protein prevails and which protein is the most abundant within LBs. Nonetheless, the assessment of alpha-syn as the major component of LBs, as perceived early on, still remains a leitmotif which starts most papers on this topic.

### 3.2. Is Alpha-Syn the Seed of LBs?

Again, present evidence does not provide any information about the dynamics of LBs concerning their formation and maturation. This needs to be clarified, since the occurrence of alpha-syn within LBs has been established so far only by non-quantitative light microscopy, and no dynamics have been investigated. It is unique that the apparent abundance of alpha-syn compared with ubiquitin was considered as a proof of principle for the specific role of alpha-syn in forming LBs. In fact, the authors stated that: “staining for α-synuclein is more extensive than staining for ubiquitin, indicating that accumulation of α-synuclein precedes ubiquitination” [78]. In our opinion, it is not consistent to establish a temporal chain of events based on a static assessment of the amounts of specific components at a specific time intervals. There is a lack of connection concerning which protein appears to be more abundant and which protein comes first. For the very same reason, one may assume the paradox that, within a family, since nephews appear to be more numerous than grandparents, they started the family and they are older. The seeding of a biological entity represents a dynamic phenomenon, which remains independent from static measurements about the abundance of various constituents at any stage.

There are additional pitfalls concerning this point. In fact, the comparison between alpha-syn and ubiquitin is likely to be inappropriate since, as previously mentioned, the occurrence of mono-ubiquitin within LBs is not relevant compared with the occurrence of poly-ubiquitin chains, which are stained by specific antibodies [77,80]. This was demonstrated by immuno-gold identification without quantitative stoichiometry in fractions obtained following the sucrose gradient centrifugation of cortical tissue from DLB patients [77]. In these specimens, various primary antibodies were tested to detect the occurrence of ubiquitin compared with that of poly-ubiquitin. The occurrence of poly-ubiquitin instead of mono-ubiquitin within LBs was firmly established [77], and this is now recognized in humans and animal models of PD [81,82,83]. In fact, in the absence of parkin activity, as occurs in PARK2 genetic parkinsonism, poly-ubiquitination operated by parkin is lost and LBs are much less abundant in these PD patients [84,85,86,87,88]. Such a critical role of parkin was demonstrated two decades ago by Schlossmacher et al. [89]. This is key, since parkin is critical for generating poly-ubiquitin chains, which, in turn, are critical for binding mitochondria and other substrates such as p62, for degradation via proteasome [90,91,92,93,94]. Such an effect is thought to be mediated through the shuttling of proteasome by the p62 sequestosome, which binds the proteasome and the poly-ubiquitin chain [95] to anchor LC3 in the phagophore. In fact, it is now widely recognized that the poly-ubiquitination of substrates is required to bind the proteasome, which, in turn, is shuttled towards the growing phagophore through an interaction with p62, allowing for the entry of the proteasome within the autophagosome compartment [96,97] (Figure 1).

### 3.3. Is Alpha-Syn the Component of Radiating Filaments within LB?

Even concerning this point, we lack an in situ immuno-gold evidence to prove such a statement. In fact, the immuno-gold staining of filaments carried out in the manuscript by Spillantini et al. [78] was obtained from non-specific brain homogenates, which were previously centrifuged. The supernatant of the centrifuged gross brain tissue was exposed for 1 h to the protease sarkosyl, and the extraction of sarkosyl-resistant filaments was carried out. An additional centrifugation of the sarkosyl-exposed supernatant led to pellets which were re-suspended and added to carbon-coated 400-mesh grids to be stained with phospho-tungstic acid and further exposed to primary anti-alpha-syn antibodies, which were revealed by immuno-gold conjugated secondary antibodies. This way, the staining of alpha-syn was not carried out in situ; thus, the source of alpha-syn-stained filaments remains uncertain concerning the types of cells and specific cell compartments. In this way, no evidence exists connecting alpha-syn-stained filaments and LBs filaments. By incidence, radiating filaments within LBs’ periphery do not seem to be often represented [1]. In fact, recent evidence toned down the classic concept of LBs as a dense halo surrounded by radiating filaments, with this structure being an exemption rather than a rule, since it was described only in one out of seventeen LBs examined under combined light and electron microscopy from PD and DLB patients [1]. Thus, to our knowledge, no evidence was provided concerning either the authentic abundance of alpha-syn within LBs nor the massive placement of alpha-syn within neurofilaments of LBs. When considering that the samples were exposed to sarkosyl, one should emphasize that the native structure of the specimen was strongly altered by the experimental procedure. In fact, phospho-tungstic acid could not stain most protein filaments, since they were previously degraded by sarkosyl. The authors assumed the analysis of the stochastically sampled brain tissue to be equivalent to the analysis of LBs [78], while immuno-gold was carried out on cell-dispersed sarkosyl-resistant filaments. Moreover, the procedure used to extract the neurofilaments was applied to original brain samples, without selecting specific cell types or subcellular compartments including LBs owing these sarkosyl-resistant filaments. Therefore, alpha-syn-immunoreactive sarkosyl-resistant filaments are likely to be stochastically distributed within various neural cells and distinct cell organelles and cytosol. In short, the evidence provided by these papers, even when using the ultrastructural study of alpha-syn filaments, was obtained from tissue samples independent from LBs. In addition, site specificity concerning brain nuclei and even the types of cells (neurons, glia, and vessels) remains non-defined. This is relevant when considering the physiologically high amount of alpha-syn within platelets, which is supposed to increase during neuroinflammation. Thus, this study did not detail the structures of filaments within LBs, since cytosolic non-LBs sarkosyl-resistant aggregated filaments are likely to be extracted instead of or in addition to LBs. In a few words, even site specificity within LBs remains unlikely and has largely not been assessed. In fact, the occurrence of alpha-syn-dense spots is commonly found in the cytosol outside of LBs within PD neurons, which may likely contribute to these filaments. Concerning this point, a recent seminal work indicated a marked anatomical and cellular discrepancy between the occurrence of LBs-related pathology and alpha-syn oligomers [98]. This is critical, since alpha-syn oligomers and even filaments are sarkosyl-resistant [99].

On the other hand, alpha-syn oligomers, which occur independently of LB pathology are key in alpha-syn-induced neurotoxicity and are also sarkosyl-resistant [100]. This makes the observation of alpha-syn immuno-gold within sarkosyl-resistant neurofilaments from dispersed brain tissue unlikely to correspond to components of LBs. Thus, the key point concerning whether the source of these filaments is within LBs or is dispersed in the cytosol remains open. The only points which remain certain concern their protein structure, the presence of alpha-syn (demonstrated by primary antibodies), and their protease resistance, since they were non-digested by sarkosyl. Their source remains undefined, being isolated from the centrifugation of homogenates from sarkosyl-resistant brain tissue, which may well apply to a number of cytosolic structures often not related to LBs [98]. This way, the experimental procedure was specifically addressed to provide a qualitative assessment of the presence of alpha-syn within brain tissue. No information was obtained concerning the occurrence of alpha-syn-positive filaments within LBs and within DA neurons of the substantia nigra. Again, the use of phospho-tungstic acid directed the analysis preferentially toward protein structures, leaving unquestioned the occurrence of other chemical species in the mesencephalon and mostly within LBs. Even the nature of the proteins within these tissue sections remained undefined, as the study aimed only to use primary anti-alpha-syn antibodies only after most native proteins were degraded by sarkosyl.

The specific evidence provided by these original studies is the occurrence of alpha-syn within LBs and the occurrence of alpha-syn within sarkosyl-resistant dispersed protein filaments from tissue homogenates from mesencephalon. However, no connection is provided between these facts.

## 4. General Doubts

To summarize the questions that still remain to be addressed before assuming an equivalence between alpha-syn as the major component of LBs, its seeding structure, and its main neurofilaments skeleton, we outline a few points here. When data are based on light microscopy immunostaining, it is difficult to establish the real abundance of a given protein, and it is quite arbitrary to compare the amount of a given protein with that of another. In fact, the signal that is produced by immunostaining revealed by peroxidase or fluorescence does not allow for protein quantification. 

Similarly, when establishing the quantitative amount of a protein from brain extracts, there is no certainty about the amount of this protein in situ within specific subcellular compartments. Therefore, only in situ quantitative stoichiometry carried out by counting immuno-gold within native, non-manipulated neuronal inclusions may allow for establishing protein amounts and comparing the amounts of various proteins within LBs [79]. Once this point is achieved, one can consider that authentic amounts of a given protein, when measured at arbitrary time intervals, do not allow for the drawing of any conclusion concerning the dynamics of forming LBs. Thus, the seeding of LBs cannot be inferred, even knowing the relative abundance of various proteins at a given time interval. Thus, even assuming that the quantitative amounts of each component of LBs are known, no conclusion about their dynamics can be firmly established yet. Therefore, the modeling of inclusions’ dynamics induced by an in tubo or in vitro artificial supply of great amounts of alpha-syn owns the inherent prejudice that alpha-syn needs to be the seed and leaves alternative mechanisms non-explored, posing some questions about whether grains of alpha-syn aggregates are indeed “bona fide model of LBs”. Instead, a timeline of the natural steps taking place during natural LBs’ formation is needed in the attempt to address such an issue thoroughly. These issues are critical, since a protein which is abundant at the late stages of a disease course does not necessarily assume a role in the seeding of LBs during the early stages. In fact, the high amount of a given protein may likely be the consequence of an impaired removal of small amounts, which slowly accumulate over time. This way, we may assume that even poorly represented proteins may be critical in the formation of an LB when their turnover is very high. The activity of these proteins may consist of seeding the inclusion and starting the process in a small time frame during a fast protein turnover, leaving in place other structures which possess a slow removal rate, though they are irrelevant in the seeding process. This concept is crucial, since it requires considering protein turnover in the context of seeding compared with the persistency of a given molecule during the natural history of LBs and PD. In fact, alpha-syn is prone to quick aggregation and protease/sarkosyl/proteinase-K resistance. In this way, it is presumable that, even when very low amounts of alpha-syn join the LB over delayed time intervals, they persist and accumulate within LBs due to a lack of degradation [16]. This is critical to consider when brain extracts are treated with protease, since, in these conditions, the presence of alpha-syn is preserved compared with that of most proteins of natural LBs, which undergo degradation. This way, the protein composition is biased by the isolation procedure. Similarly, feeding in tubo systems or in vitro cell lines with cargoes of mutated, non-digestible proteins carries a one-way hypothesis concerning the guilty seeding protein, with no chance to analyze the natural time course and the whole scenario. There is no reason to infer that the abundance of alpha-syn represents a proof of principle either that its accumulation precedes ubiquitination or that its accumulation occurs earlier than ubiquitin (indeed poly-ubiquitin). As a matter of fact, it is more likely that the degradation of one protein is slower than the other, which, in fact, is the case. Accordingly, the resistance of alpha-syn to protease is likely to generate the persistency of this protein in the tissue, even when its generation is slow in time and/or low in amount. It is likely that, in condensed time intervals, the presence of alpha-syn may be negligible compared with p62, which suggests a limited role in the seeding mechanism, as demonstrated ex vivo [13,16]. Despite these questionable issues, conclusions have been drawn in most articles that alpha-syn represents the most abundant component of LBs and concomitantly the earliest building block in LBs’ formation. Similarly, the apparent abundance of alpha-syn within dispersed cell filaments has started a stream of beliefs that LBs should be naturally made up of alpha-syn peripheral radiating filaments, while present evidence in situ suggests that the shell of LB is, rather, produced by mitochondria and vesicular membranes [1].

## 5. Alternative Mechanisms

The assumption of original research works started from the concept that LBs are made up of protein filaments. In fact, the starting concept was that LBs were made of abnormal filamentous assemblies of an unknown composition. Indeed, even pioneer works sustained this structural composition of LBs. For instance, Duffy and Tennyson [101] and Roy and Wolman [102] defined LBs as being made up by filaments both in the core and the shell. This assumption is now modified, since it is now evident that LBs are, rather, made up of dysmorphic organelles, among which, mitochondria and endo-lysosomes prevail [1]. In keeping with original concepts, alpha-syn was recognized in the periphery of LBs and LBs were defined an assembly of filaments. It is very likely that, within a PD neuron, alpha-syn-stained filaments may occur mostly distinct from LBs. In fact, alpha-syn immunostaining is abundant in these neurons also in cell domains, which do not contain LBs. The profuse accumulation of alpha-syn in PD was recognized independently of LB formation by a number of studies [98,103]. This is a crucial point suggesting that some confusion exists between LB-related pathology and alpha-syn pathology, since they may often relate to different morphological and pathophysiological entities [98]. 

Amongst alpha-syn-stained dispersed filaments, various shapes are evident, including slender filamentous structures, small clumps of filaments associated with amorphous material. The uncertainties about the connection between alpha-syn pathology (mostly sustained by alpha-syn oligomers) and LB pathology came into focus with the manuscript by Sekiya and colleagues [98], who used a proximity ligation assay (PLA) to detect dispersed alpha-syn oligomers within neurons [98]. These authors, by using PLA, demonstrated a marked discrepancy between the occurrence of alpha-syn oligomers and LB pathology in PD at the level of the substantia nigra and locus coeruleus. These correspond to the brainstem nuclei known to be mostly affected (to a similar severe extent) in PD, which develop a slighter presence of alpha-syn oligomers in the presence of severe LB pathology in PD patients. These findings suggest that sampling dispersed alpha-syn filaments is unlikely to provide any significant information concerning the role, structure, and even the occurrence of alpha-syn within LBs. In line with this, a reversed discrepancy is shown by PLA at the cortical level (crucial in DLB), where the occurrence of LB pathology is slighter compared with the severe burden of alpha-syn oligomers [98]. The inference that alpha-syn is key in forming LB was further challenged by a mathematical simulation based on updated LB composition by [11,12]. In these studies, the progression of LB could be induced by aggregates of lipid membrane-containing organelles. As stated by these authors, according to the model, the formation of LBs is catalyzed by aggregates of membrane-bound organelles, even in the absence of alpha-syn fibrils, as reported in the introduction. In fact, a great limitation of current studies based on the dynamics of LBs as derivatives of progressive alpha-syn aggregation is biased by the specificity of biological samples. These concepts pose the question of whether alpha-syn or even other proteins may be guilty by an association rather than initiating the seeding of LBs. Thus, alpha-syn may be guilty by association or even represent innocent by-standers [7,104].

### 5.1. The Good and the Bad Copes of Alpha-Syn

The persistence of alpha-syn compared with other proteins at prolonged time intervals is likely to be inherent to the trend of alpha-syn aggregating and developing non-digestible structures. This is not the case for p62 and poly-ubiquitin. Indeed, present evidence indicates that both p62 and poly-ubiquitin generate inclusions at early time intervals, before alpha-syn is added. Thus, the dynamics of inclusions’ formation may, in turn, occur according to the opposite timing concerning alpha-syn, p62, and poly-ubiquitin in the process of forming LBs. In fact, if we consider that both poly-ubiquitin and p62 may be easily digested, while alpha-syn is not, then it is likely that the amounts of p62 and poly-ubiquitin that are abundant at early and late time intervals should be multiplied for the reciprocal of their half-life, making it likely that thousands of p62 or poly-ubiquitin proteins participate in forming LBs compared with a few alpha-syn molecules. The latter may be added, once in a while, and persist as quite inert non-digestible matter. As a paradox opposed to current beliefs, the abundant immunostaining of LB for alpha-syn may be simply the effects of the irreversible accumulation of a protein, which is not so critically involved in the pathophysiology of either inclusions or sporadic PD. In contrast, alpha-syn in sporadic PD may exert a beneficial effect. In fact, alpha-syn may partly counteract the ongoing degeneration. As expected by the high complexity of alpha-syn neurobiology, this protein possesses both good and bad copes, depending on the context. Therefore, in addition to detrimental effects, alpha-syn may also have beneficial effects. In fact, the physiological role of alpha-syn as a chaperonin is able to compensate for the loss of other protective proteins [105]. Similarly, the presence of alpha-syn protects against methamphetamine-induced nigrostriatal DA toxicity [106] and against toxicity produced by DA in the absence of parkin [107]. Similarly, the evolvability of normally conformed alpha-syn may counteract some detrimental steps in the cytopathology of PD [108]. In the physiology of the cell, alpha-syn is seminal for endocytosis [109] and for providing membrane conformations, and it is pivotal in producing membrane bending [110]. In fact, the physiological regulation of the presynaptic terminals at the level of the nigrostriatal system can involve all three synucleins belonging to the alpha-syn family [111]. Therefore, alpha-syn may be not a detrimental, but rather a compensatory protein in sporadic PD, where its role as a biomarker relates to a chaperonin-like compensatory effect. In these cases, the occurrence of increased alpha-syn within affected cells may be the consequence of altered protein removal, due to a primary defect in the protein-clearing system. This is in sharp contrast with rare familial autosomal PD, which is produced by alpha-syn gene point mutations or multiplications, as reported in paragraph 1.5.

### 5.2. The Good and the Bad Copes of LBs 

Although the occurrence of LBs is considered, per se, as a hallmark of PD degeneration, their intimate significance remains to be established. In fact, they may just represent: (i) the inert accumulation of otherwise cytotoxic molecules, which are less harmful when being aggregated; (ii) evidence for the activation of a compensatory response to counteract detrimental phenomena (which would explain the occurrence of chaperone proteins within LBs); (iii) evidence for a failure to metabolize protein cargoes, altered mitochondria, and an altered retromer/endosome turnover; or (iv) the co-existence of all these features. In addition, the presence of LBs may inherently confer toxicity to the cell either as mechanic disruptors of cell metabolism or even indirectly by triggering neuroinflammation. This phenomenon appears to play a significant role in PD [112]. In fact, LBs, as shown recently [113,114], foster the occurrence of neuroinflammation in PD and high IgG levels correlate with the amounts of LBs in PD patients. Within the context of LBs-induced neuroinflammation, most manuscripts investigated the pro-inflammatory role of alpha-syn, which is in line with the prominent role currently attributed to alpha-syn within LBs. Nonetheless, as shown by other reports, a number of protein/chemical species included within LBs, under the effects of a detrimental amount of free radicals and oxidative species, may, in turn, trigger neuroinflammation [115].

## 6. The Hypothetical Role of p62 and Poly-Ubiquitin in Seeding Vesicles-Rich Inclusions 

Recent data based on LBs’ composition indicate a large amount of membrane vesicles derived from the autophagosome, endosome, and lysosome compartments and their various merging structures (Figure 1 and Figure 2), along with a dense clustering of damaged mitochondria in the periphery. When this evidence is joined with the molecular genetics of parkinsonism and the cell pathology of PD, novel pathogenic indications occur. In fact, a dysfunctional membrane vesicle compartment started either in the retromer compartment of endosomes or within altered clathrin-mediated endocytosis [4,20] may be represented by engulfing multivesicular bodies (MVB). This may be added to or alternative to an early dysfunction of the autophagy machinery both at early and late stages, impeding the merging with lysosomes [30,50,52,53,54,56,60,116,117,118,119,120,121,122]. Still, in this scenario, a critical step involves the shuttling of the proteasome via p62 towards the nascent phagophore [123], where poly-ubiquitin chains interact with p62 to deliver misfolded proteins to segregated domains of degradation. Within this frame, altered degradation constantly involves the mitochondrial compartment, which was constantly found to be altered in PD patients [124,125]. It is very likely that alterations of all these structures may start the disease process. Nonetheless, it seems that, independently from the primary affected step, all trafficking is impaired when disease progresses. This explains the constancy of many cyto-pathological alterations (altered mitochondria, stagnant autophagosomes, misfolded proteins accumulation, altered MVB, and relented endosomal progression), which are quite constant in PD diseased neurons. In line with this, the biochemical measurement of mitochondrial activity, along with lysosomal efficacy and proteasome efficiency, are all altered. Consistently, when mutations occur in those genes, which regulate proteasome, autophagy, lysosomes, mitochondria, or key proteins in cargo delivery, genetic parkinsonism is produced [126,127]. These multi-step scenarios should be considered in the effort to comprehend PD and LB seeding and maturation. Here, we wish to emphasize the impressive findings concerning the amount of p62 and poly-ubiquitin clusters, which occur in humans and experimental models of PD. Somehow, this suggests that, no matter where the occlusion of cell clearance occurs, an upstream engulfment of the buffering ability of p62 and poly-ubiquitin occurs as well. This impairs the driving of misfolded proteins and altered mitochondria to the appropriate sites of degradation and may represent the template of PD cell pathology. Dedicated studies are needed to decipher the subtle balance between p62, poly-ubiquitin, proteasome, and autophagosomes (as shown in Figure 1 and Figure 2) as critical points in the dynamics of LBs’ seeding and progression. This vision rules out a single protein-centered approach to comprehending the genesis of LBs and PD. It rather emphasizes a few organelles and their related shuttling proteins as initiators of LBs’ formation. We would not be surprised if alpha-syn, apart from rare genetic mutations, would turn out to be an innocent protein involved in chaperone-related functions, which inevitably accumulates due to a slow aggregation based on its native, non-digestible conformation. This could be the key for the constant presence of such a protein, which may be overemphasized concerning its seeding role, total amount, and site specificity for LBs, as we discussed in the various paragraphs of the present manuscript.

## Figures and Tables

**Figure 1 ijms-25-03929-f001:**
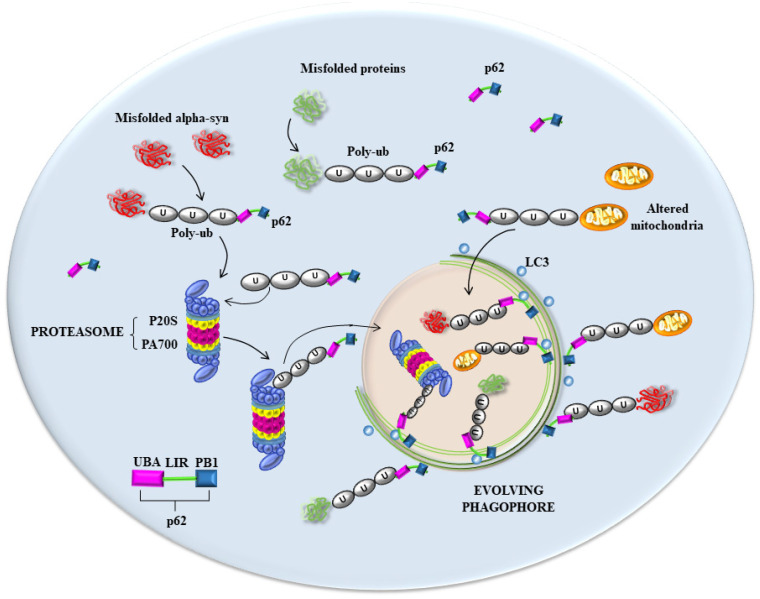
A hypothesis about the seminal role in PD of key proteins involved in shuttling the proteasome into the phagophore. In the process of removal of unfolded and misfolded proteins, as well as mitochondria, multiple clearing pathways appear to converge [97]. As shown in the scheme, both poly-ubiquitinated proteins (including alpha-syn) and mitochondria are tagged by p62, which works as a shuttle to deliver these substrates to LC3-stained structures such as the autophagosome [96]. The transfer of these substrates to nascent phagosomes is likely to be impaired in PD, which generates the accumulation of clustered amounts of p62 and poly-ubiquitin close to non-effective autophagosomes, omegasomes, and a number of misfolded proteins, including alpha-syn, which cannot be properly degraded. The role of p62 is pivotal in connecting poly-ubiquitin, misfolded proteins, proteasome, and nascent phagophore. This occurs since p62 owns three key domains: (i) ubiquitin-associated domain (UBA), which binds poly-ubiquitin chains; (ii) LC3-interactive region (LIR), which binds LC3, allowing for its anchoring to phagosomal membranes and all membranes potentially evolving as phagosomes, including those arising from the trans-Golgi network (TGN); and (iii) Phox and Bem1p (PB1) domains, which allow binding to the proteasome. This explains why p62 and poly-ubiquitin cluster together in great amounts close to phagosomal membranes when autophagy is inhibited [31]. Similarly, this may explain the great amount of clustered p62 and poly-ubiquitin at the level of eosinophilic LB-like structures, which are generated following proteasome inhibition [13,14] and during expression of PD-related genes [15,16,17].

**Figure 2 ijms-25-03929-f002:**
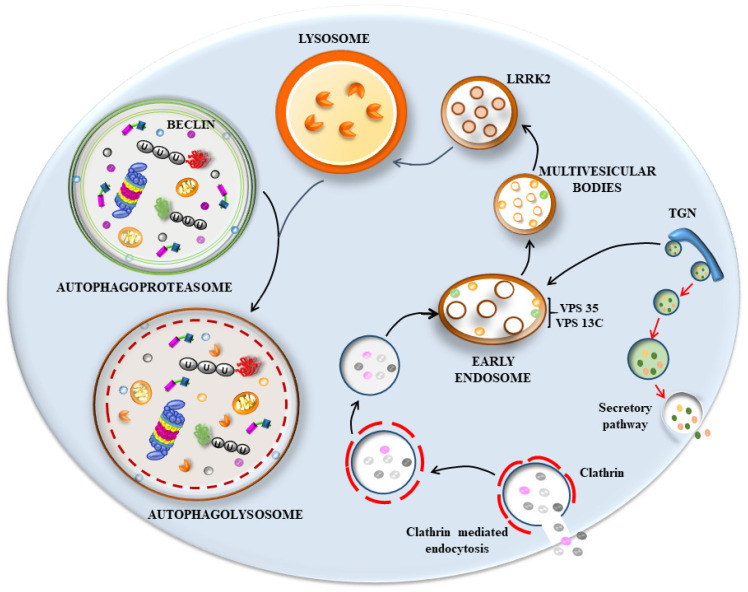
Scheme connecting the retromer and endosome pathway to autophagy in producing an impairment of endolysosomal clearance in PD. Hypothetical scenario on the right part of the cartoon shows dysfunctional membrane vesicle compartments started either in the retromer component of endosomes or within altered clathrin-mediated endocytosis [4,20]. This may lead to engulfing multivesicular bodies (MVB). This may be added to or alternative to early dysfunction of the autophagy machinery (left part of the cartoon, and Figure 1) both at early and late stages, impeding the merging of the endosome/autophagosome system with lysosomes [30,50,52,53,54,56,60,116,117,118,119,120,121,122]. LRRK2: leucine-rich repeat kinase is altered in a genetic form of PD and it is involved in endolysosomal pathway. Specifically, it regulates endocytosis, vesicle-trafficking pathways, and lysosomal degradation [128]. The activity of LRRK2 is at the intersection of other proteins involved in PD and endosomal retromer pathway such as vacuolar protein sorting 35 (VPS35) and 13c (VPS13c). Mutations of VPS35 and VPS13c alter the trafficking of retromers and cause PD [52,129]. Endosomes are mainly formed at the level of the trans-Golgi network (TGN) and move either towards plasma membrane to secrete cargoes or move towards the endosomal complex. At this level, they merge with endocytic vesicles derived from the endocytosis of clathrin-coated vesicles and progress to multivesicular bodies (MVB) for lysosomal degradation. This latter step may occur selectively for endosomes and late endosomes (MVB) or following the merging of endosomes with autophagosomes (in the left part of the cartoon) to form amphisomes, which ultimately enter the lysosomes. Since most, if not all, of these vesicle compartments can be formed by similar intracellular LC3-stained membranes, including some of the trans-Golgi network (TGN), it is likely that an impediment of one of these compartments that eventually leads to a downstream cascade affecting all these organelles.

## Data Availability

Not applicable.

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
