# Peer review of "Is There a Place for Lewy Bodies before and beyond Alpha-Synuclein Accumulation? Provocative Issues in Need of Solid Explanations"

_ijms, 2024, doi:10.3390/ijms25073929_

Round 1
Reviewer 1 Report
Comments and Suggestions for Authors
This is a timely and important review article. Alpha-synuclein (alpha-syn) has been assumed to be a major component of Lewy bodies (LBDs) and to play an important role in forming peripheral leading filaments as an early biomarker of Parkinson's disease (PD). The present article in keeping with a signigficant role of alpha-syn with LBDs peresents a provocative issue challenging dogmas about a prominent role of alpha-syn compaired with other proteins or non-protein molecules. The present discussion is based on several recent controvetial references, such as Reference [67] Sekiya et al. (Discrepancy between distribution of alpha-synuclein oligomers and Lewy body-deduced pthology in Paekison's disease. Acta Neuropthol. Commun, 2022). The authors pointed out that there have been no confirmation by solid methods on the presence of alpha-syn with LBDs, such as immuno-gold quantification [Ref. 48]. The specific evidence was the occurence of alpha-syn within LBDs and the occurence of alpha-syn within sarcosyl-resistant dispersed protein filament from tissue homogenate from mesencephalon , but no connection was present between these two.
The present reviewer has the following additonal comments.
1) "LBDs formation is an autocatalytic process." Please explain this sentence in more deail by citing references.
2) What do authors consider on the physisiological and pathologicl role of alpha-syn in connection on this provocative issue, i.d., early biomarker of PD?
3) Please explain the present proposl in relation to the recent findings on mutations in the genes of genetic PD, which regulate proteasome, autophagy, lysosomes, mitochondria, and key proteins in cargo delivery.
Author Response
Reviewer#1
General comment
This is a timely and important review article. Alpha-synuclein (alpha-syn) has been assumed to be a major component of Lewy bodies (LBDs) and to play an important role in forming peripheral leading filaments as an early biomarker of Parkinson’s disease (PD). The present article in keeping with a significant role of alpha-syn with LBDs peresents a provocative issue challenging dogmas about a prominent role of alpha-syn compaired with other proteins or non-protein molecules. The present discussion is based on several recent controvetial references, such as Reference [67] Sekiya et al. (Discrepancy between distribution of alpha-synuclein oligomers and Lewy body-deduced pthology in Paekison’s disease. Acta Neuropthol. Commun, 2022). The authors pointed out that there have been no confirmation by solid methods on the presence of alpha-syn with LBDs, such as immuno-gold quantification [Ref. 48]. The specific evidence was the occurrence of alpha-syn within LBDs and the occurrence of alpha-syn within sarcosyl-resistant dispersed protein filament from tissue homogenate from mesencephalon , but no connection was present between these two.
Response
We wish to thank the reviewer for finding merits in our work and considering it as a timely and important review article.
The present reviewer has the following additional comments.
- "LBDs formation is an autocatalytic process." Please explain this sentence in more deail by citing references.
Response
We explained the sentence in the revised version and we referred to specific reference: LBs formation is often considered an autocatalytic process, which means that the early seed of LB further progresses into mature LB formation autonomously, through self-assembly and interactions of various components due to their chemical properties. In fact, as reported by recent studies (Kuznetsov and Kuznetsov 2022 Kuznetsov 2024) LB formation is an autocatalytic process, where progression of LBs formation will be catalyzed by aggregates of membrane-bound organelles even in the absence of α-syn aggregates.
- What do authors consider on the physisiological and pathologicl role of alpha-syn in connection on this provocative issue, i.d., early biomarker of PD?
Response
The reviewer is right to rise physiological and pathological functions of alpha-syn. As expected by the high complexity of neurobiology alpha-syn possesses both good and bad cops depending on the context. Therefore, we added on the concept that alpha-syn may have either beneficial or detrimental effects: In fact, the physiological role of alpha-syn as a chaperonin being able to compensate for a loss of other protective (chaperon) proteins was reported (Chandra et al., 2005). Similarly, the presence of alpha-syn appears to protect against specific toxic conditions such as methamphetamine toxicity (Schlüter et al., 2004) and DA toxicity in the absence of parkin (Machida et al., 2005). Similarly, the evolvability of normally conformed alpha-syn may counteract some detrimental steps in the cytopathology of PD (Wei et al., 2021). In the physiology of the cell, synuclein is seminal for endocytosis (Vargas et al., 2014) and for providing membrane conformations and it is pivotal in producing membrane bending (Westphal and Chandra, 2013). Indeed, the physiological regulation of the presynaptic terminals at the level of the nigrostriatal system can involve all three synucleins belonging to the alpha-syn family (Anwar et al., 2011). The pathological effects of alpha-syn were reported concerning both point mutations and gene multiplications, which are responsible for rare cases of genetic Parkinsonism. These pathological effects were mostly reported in relationship with the deleterious effects of mutated or massive amounts of alpha-syn in regulating proteasome activity (Fornai et al., 2005; McKinnon et al., 2020) and autophagy activity (Nascimento et al., 2020) which is backed up also in the response to the following reviewer’s point at point 3. Therefore, to address the key question of the reviewer alpha-syn may be not a detrimental but rather a compensatory protein in the vast majority of PD cases, where its role as a biomarker relates to a chaperonin-like effect. In these cases, the occurrence of increased alpha-syn within affected cells may be the consequence of altered protein removal, due to a primary defect in protein clearing system. In contrast, in rare genetic PD cases characterized by point mutations or multiplications in the alpha-syn gene, the protein alpha-syn becomes toxic. In fact, mutated and multiple alpha-syn are shown to be toxic (deleterious) on major clearing pathways such as proteasome and autophagy as reported and referenced above. In these rare inherited conditions, alpha-syn may become fundamental in generating the neurobiology of disease being the primary detrimental cause to alter autophagy and proteasome activity.
- Please explain the present proposl in relation to the recent findings on mutations in the genes of genetic PD, which regulate proteasome, autophagy, lysosomes, mitochondria, and key proteins in cargo delivery.
Response
This is in line and it repurposes the previous point. In fact, this relationship was added including the evidence about genetic PD affecting mutations in genes, which regulate cell clearing pathways such as autophagy, proteasome, lysosome, which are seminal for mitochondrial turnover and delivery of protein cargoes. Appropriate references were included. In particular, we further detailed the commonalities about cell pathology in PD starting from recent finding about specific mutations. In this way mutated alpha-syn (either due to point mutations or gene multiplications produces the impairment in autophagy and proteasome, while most cases of PD would feature an alpha-syn independent impairment of cell clearance. In this case the accumulation of non-mutated alpha syn would occur as innocent bystander or as a guilty element only by association. In fact, a perturbation protein clearance would sort an accumulation of alpha-syn, which in turn further inhibits both proteasome and autophagy (Zalon et al., 2024). On the other hand, while physiological alpha-syn is a substrate of autophagy, mutated alpha-syn is often resistant to autophagy degradation (Matsuki et al., 2024). This point is further addressed in the final part of the manuscript, where genetic defect at specific steps in cell clearing systems (autophagy, endosomes and proteasome) were related to abnormal persistence of altered mitochondria and ineffective removal of protein cargoes (including alpha-syn). These mutations may primarily involve different interconnected compartments such as autophagosomes, proteasome, endosomes, retromers, lysosomes, mitochondrial turnover, primary protein misfolding.

Reviewer 2 Report
Comments and Suggestions for Authors
This is a provocative review of Lewy body (LB) formation in Parkinson's disease (PD). I support this type of analysis, which is healthy for the PD community, as evolving dogma needs to be questioned. What the authors have done ultimately is to question the primacy of alpha-synuclein protein (ASP) as the main driver for LB formation in PD and related conditions. They incorporate multiple other ubiquitinated intracellular components into LB's (e.g. proteasome, mitochondria) when p62 "sequestome" protein leads to the formation of LB intracellular inclusions that clearly contain ASP but probably do not arise solely from ASP. This is an important concept, as ASP is considered by some investigators as the inciter of PD pathology and the major cause of PD pathogenesis, such that some experimental therapeutics for PD are directed at removing ASP from the brain. If ASP is only one of several components of LB, as the authors suggest, then this approach is necessarily limited.
No one would deny that LB's contain ASP; but the question explored by the authors relates to the problem that, quantitatively, ASP has not ever been shown to be the dominant protein of LB's. The authors present studies that suggest past attempts to address this question are flawed. This lack of quantitative data must be considered in light of the knowledge that a). one rare form of PD appears to arise from multiple copies of the ASP gene, and b). rare autosomal dominant forms of PD arise from point mutations in the ASP gene.
One major deficit of the paper is that the authors do not discuss or even mention that there are rare autosomal causes of PD that appear to involve either extra copies of ASP gene or mutations in ASP gene. This deficit is heightened by their mentioning other genetic causes of PD.
The authors do an excellent job of showing that although ASP protein is present in LB's, it may not be the sole or even main driving force behind LB formation. As mentioned previously, there has been an evolving dogma that "LB's define PD, LB's contain ASP, therefore ASP causes PD". The authors indirectly and gently debunk this dogma and point out that ASP tends to form fibrils spontaneously, is present in fibrillar form outside of LB's, and thus that injection of ASP fibrils into brain is not really a good model for spontaneous PD (the vast majority of cases).
The authors do not mention the evolving idea that PD and other neurodegenerations appear to arise from consequences of activating one or more inflammation pathways. Theoretically they do not have to mention this concept in their review, but I feel their review will benefit from the proposal that LB's (which they show are composed of many things beyond ASP) could stimulate such a response.
Although I enjoyed their provocative arguments, I do feel that their paper would benefit from editing. Their English is overall fine, with a few dangling subject-verb problems that are easily corrected. The major deficit to me is organization. Their paragraphs are too long and present too many different concepts. Perhaps they could try subheadings to present their intriguing ideas more clearly.
Overall I feel this is an important, likely controversial and timely review. With careful editing, I feel that it will have a larger impact. It is important that the "ASP dogma" of PD pathogenesis be challenged, and I think the authors are on one of several right tracks here.
Comments on the Quality of English Languagesee above
Author Response
Reviewer#2
General comment
This is a provocative review of Lewy body (LB) formation in Parkinson's disease (PD). I support this type of analysis, which is healthy for the PD community, as evolving dogma needs to be questioned. What the authors have done ultimately is to question the primacy of alpha-synuclein protein (ASP) as the main driver for LB formation in PD and related conditions. They incorporate multiple other ubiquitinated intracellular components into LB's (e.g. proteasome, mitochondria) when p62 "sequestome" protein leads to the formation of LB intracellular inclusions that clearly contain ASP but probably do not arise solely from ASP. This is an important concept, as ASP is considered by some investigators as the inciter of PD pathology and the major cause of PD pathogenesis, such that some experimental therapeutics for PD are directed at removing ASP from the brain. If ASP is only one of several components of LB, as the authors suggest, then this approach is necessarily limited.
Response
We wish to thank the reviewer for considering the manuscript as a kind of analysis which is healthy for the PD community and for his thoughtful comments we fully share.
No one would deny that LB's contain ASP; but the question explored by the authors relates to the problem that, quantitatively, ASP has not ever been shown to be the dominant protein of LB's. The authors present studies that suggest past attempts to address this question are flawed. This lack of quantitative data must be considered in light of the knowledge that a). one rare form of PD appears to arise from multiple copies of the ASP gene, and b). rare autosomal dominant forms of PD arise from point mutations in the ASP gene. One major deficit of the paper is that the authors do not discuss or even mention that there are rare autosomal causes of PD that appear to involve either extra copies of ASP gene or mutations in ASP gene. This deficit is heightened by their mentioning other genetic causes of PD.
Response
The reviewer is right in fact as suggested also by reviewer 1 we mentioned the cases where point mutations or multiplications of alpha-syn gene produce rare forms of genetic PD. This point was discussed in the light of the effects of alpha-syn as a substrate for proteasome and autophagy and its potential deleterious effects on these seminal clearing pathways when the gene is mutated or following gene multiplications. In fact, evidence is provided and appropriately reference that rare and severe phenotypes of PD are induced by multiplication of the alpha syn gene (Singleton et al., 2003; Farrer et al., 2004; Ferese et al., 2015). which are shown to impede the natural mechanisms of cell clearance (Nascimento et al., 2020). Similarly, point mutations of alpha syn may produce various phenotypes (depending on the site of the mutation) of PD being powerful inhibitor of cell clearing pathways such as proteasome (Jiang et al., 2010; McKinnon et al., 2020) and autophagy (Yan et al., 2014; Winslow and Rubinsztein, 2011; Winslow et al., 2010). In detail, overexpression of alpha-synuclein disrupts the placement and the move of Atg9 from trans Golgi network towards LC3-positive vesicles, which determines a loss of autophagosomes and impaired autophagy, which is constant in PD pathogenesis. This points also answer the issue of the bad and good cops of alpha-syn in the cell. This is addressed following the subsequent Reviewer’s comment which insists in this topic.
The authors do an excellent job of showing that although ASP protein is present in LB's, it may not be the sole or even main driving force behind LB formation. As mentioned previously, there has been an evolving dogma that "LB's define PD, LB's contain ASP, therefore ASP causes PD". The authors indirectly and gently debunk this dogma and point out that ASP tends to form fibrils spontaneously, is present in fibrillar form outside of LB's, and thus that injection of ASP fibrils into brain is not really a good model for spontaneous PD (the vast majority of cases).
Answer/Comment
Yes, this reviewer comments covers our intent.
The authors do not mention the evolving idea that PD and other neurodegenerations appear to arise from consequences of activating one or more inflammation pathways. Theoretically they do not have to mention this concept in their review, but I feel their review will benefit from the proposal that LB's (which they show are composed of many things beyond ASP) could stimulate such a response.
Response
Yes, we understand and empathize with the Reviewer’s intent of mentioning such an issue.
Therefore, we mentioned the potential role of LBs as potential stimulators of neuroinflammation.
Therefore, we added appropriate sentences in the revised version. The significance of LBs: Although the occurrence of LBs is considered per se as a hallmark of PD degeneration, their intimate significance remains to be established. In fact, they may just represent: (i) the inert accumulation of otherwise cytotoxic molecules which are less harmful when being aggregated; (ii) the evidence for the activation of a compensatory response to counteract detrimental phenomena (which would explain the occurrence of chaperone protein within LBs); (iii) the evidence for a failure to metabolize protein cargoes, altered mitochondria and an altered retromer /endosome turnover; (iv) the co-existence of all these features. In addition, the presence of LBs may inherently confer toxicity to the cell either as mechanic disruptor of cell metabolism or even indirectly by triggering neuroinflammation, which appear to play a significant role in PD (Riederer et al., 2023). In fact, LBs as shown recently (Murakami et al., 2023) occurrence of neuroinflammation in PD and high IgG levels correlates with amount of LBs in PD patients. Within the context of LBs-induced neuroinflammation most manuscripts investigated the pro-inflammatory role of alpha-syn, which is in line with the prominent role currently attributed to alpha-syn within LBs. Nonetheless, as reported by other reports, a number of protein/chemical species included within LBs, under the effects of a detrimental amount of free radicals and oxidative species, may turn to trigger neuroinflammation (Sian-Hulsmann et al., 2015)
Although I enjoyed their provocative arguments, I do feel that their paper would benefit from editing. Their English is overall fine, with a few dangling subject-verb problems that are easily corrected. The major deficit to me is organization. Their paragraphs are too long and present too many different concepts. Perhaps they could try subheadings to present their intriguing ideas more clearly.
Response
Thank you for enjoying the manuscript. We provided extensive editing mostly dedicated to express separate concepts and provide subheadings and a clear organization mostly in the long introduction which is now splitted into 5 subheadings and novel subheadings in paragraph 3 and paragraph 4, as the reviewer suggested.
Overall I feel this is an important, likely controversial and timely review. With careful editing, I feel that it will have a larger impact. It is important that the "ASP dogma" of PD pathogenesis be challenged, and I think the authors are on one of several right tracks here.
Response
Thank you very much indeed for these positive comments

Round 2
Reviewer 2 Report
Comments and Suggestions for Authors
This revised version addresses all of my concerns and even more strongly addresses the contemporary prejudice about the role of ASN in PD. I support its publication once the minor English grammar problems are addressed. I note there have been substantial additions and revisions, and the authors have modified (improved) the flow of the paper substantially.
Comments on the Quality of English LanguageAs in the original version, the English grammar overall is fine. There are a few lingering subject-verb mismatches that are easily corrected.